# Boosting Asynchronous Decentralized Learning with Model Fragmentation

## Abstract

Decentralized learning (DL) is an emerging technique that allows nodes on the web to collaboratively train machine learning models without sharing raw data. Dealing with stragglers, *i.e.*, nodes with slower compute or communication than others, is a key challenge in DL. We present DivShare, a novel asynchronous DL algorithm that achieves fast model convergence in the presence of communication stragglers. DivShare achieves this by having nodes fragment their models into parameter subsets and send, in parallel to computation, each subset to a random sample of other nodes instead of sequentially exchanging full models. The transfer of smaller fragments allows more efficient usage of the collective bandwidth and enables nodes with slow network links to quickly contribute with at least some of their model parameters. By theoretically proving the convergence of DivShare, we provide, to the best of our knowledge, the first formal proof of convergence for a DL algorithm that accounts for the effects of asynchronous communication with delays. We experimentally evaluate DivShare against two state-of-the-art DL baselines, AD-PSGD and Swift, and with two standard datasets, CIFAR-10 and MovieLens. We find that DivShare with communication stragglers lowers time-to-accuracy by up to 3.9× compared to AD-PSGD on the CIFAR-10 dataset. Compared to baselines, DivShare also achieves up to 19.4% better accuracy and 9.5% lower test loss on the CIFAR-10 and MovieLens datasets, respectively.

## CCS Concepts

• **Computing methodologies → Distributed artificial intelligence**; **Distributed algorithms**.

## Keywords

Decentralized Learning, Collaborative Machine Learning, Asynchronous Decentralized Learning, Communication Stragglers

**ACM Reference Format:**
Anonymous Author(s). 2024. Boosting Asynchronous Decentralized Learning with Model Fragmentation. In . ACM, New York, NY, USA, 14 pages. https://doi.org/10.1145/nnnnnnn.nnnnnnn

## 1 Introduction

Decentralized learning (DL) is a collaborative learning framework that allows nodes on the web to train a machine learning (ML) model without sharing their private datasets with others and without the involvement of a centralized coordinating entity (*e.g.*, a server) [38]. During each round of DL, nodes independently train their models using their private dataset. Based on a specified communication topology, the updated local models are then exchanged with *neighbors* over the Internet and aggregated at each recipient node. The aggregated model serves as the starting point for the next round, and this process continues until convergence. This approach enables web-based applications, such as recommender systems [6, 13, 40] or social media [10, 27], to collaboratively leverage the capabilities of ML models in a privacy-preserving and scalable manner. Notable DL algorithms include Asynchronous decentralized parallel stochastic gradient descent (AD-PSGD) [39], Gossip learning (GL) [47], and Epidemic learning (EL) [12].

It is natural for nodes in any real-world network to have different computation and communication speeds. While the focus on DL has been surging due to its wide range of applicability [7], most existing works in DL consider a synchronous system without the presence of *stragglers*, *i.e.*, nodes with slower compute or communication speeds than others [9, 12, 18, 39, 43, 54]. In synchronous DL approaches such stragglers can significantly prolong the time required for model convergence as the duration of a single round is typically determined by the slowest node [38]. Ensuring quick model convergence in the presence of stragglers and reducing their impact is crucial to improving the practicality of DL systems.

This work deals with *communication stragglers* in DL. This form of system heterogeneity is particularly present in web-based systems where nodes are geographically distributed and inherently have variable network speeds, resulting in delayed communications [20]. For instance, the network speeds of web-connected mobile devices can differ by up to two orders of magnitude [35]. Even when nodes are deployed over cross-region AWS instances, their network bandwidth can vary by up to 20× [20].

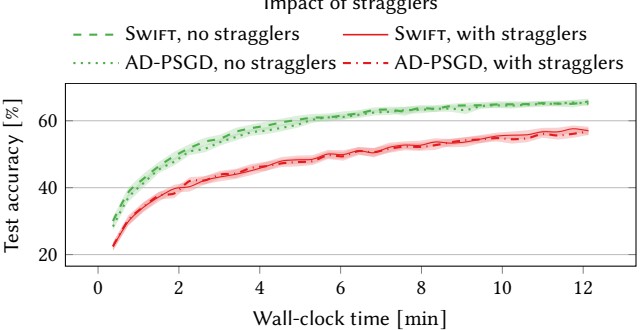

**Figure 1: The convergence plots for AD-PSGD and Swift on CIFAR-10, with and without communication stragglers.**

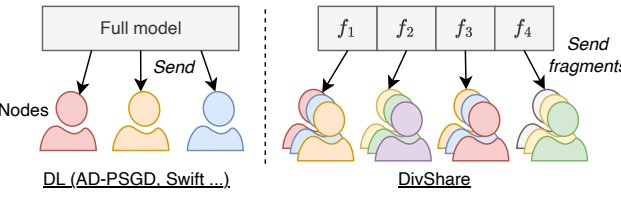

**Figure 2: Model sharing in DL (left) and DivShare (right), from the perspective of a single node. DivShare fragments models and sends each fragment to randomly selected nodes.**

In the context of DL, this variability in communication speed results in slower model convergence. In most DL algorithms, a node sends its full model to a few other nodes (Fig. 2, left). Nodes with slow network links require more time to transfer larger models, which can lead to two negative outcomes for their parameter updates: *(i)* they are received later and become stale by the time they are merged, or *(ii)* they are ignored entirely as the recipient proceeds with aggregation without waiting for the contributions of slow nodes. Additionally, from the perspective of a sending node with a fast network link, a slow recipient can block valuable bandwidth. In many DL algorithms, a recipient only proceeds with aggregation after receiving the full model. Rather than communicating with faster nodes, the sender must wait, further exacerbating the straggling effect and hindering system performance.

We highlight the impact of communication stragglers in DL with an experiment where we measure the model convergence of AD-PSGD [39] and Swift [9], two state-of-the-art asynchronous DL algorithms, in a 60-node network. We consider an image classification task with the CIFAR-10 dataset [34] and use a non independent and identically distributed (non-IID) data partitioning [14, 42]. Fig. 1 shows the test accuracy of AD-PSGD and Swift over time, without communication stragglers (green curves) and with communication stragglers (red curves) where half of the nodes have 5× slower network speeds than others. For both AD-PSGD and Swift, we observe a significant reduction in model convergence speeds. To reach 56 % test accuracy, AD-PSGD and Swift require 1.9× and 2.2× more time, respectively, in the presence of communication stragglers compared to when they are absent.

To address this issue, we introduce DivShare: a novel asynchronous DL algorithm. DivShare converges faster and achieves better test accuracy compared to state-of-the-art baselines in the presence of communication stragglers. Specifically, after finishing their local training (computation), nodes in DivShare *fragment* their models into small pieces and share each fragment independently with a random set of other nodes (Fig. 2, right). Transferring smaller fragments allows nodes with slow network links to quickly contribute at least with some of their model parameters. Furthermore, the independent dissemination of fragments to random sets of nodes enables more efficient use of the collective bandwidth as model parameters reach a bigger stretch of the network. As a result, DivShare exhibits stronger straggler resilience and attains higher model accuracy compared to other asynchronous DL schemes.

**Contributions.** Our work makes the following contributions:

(1) We introduce DivShare, a novel asynchronous DL approach that enhances robustness against communication stragglers by leveraging model fragmentation (Sec. 3).

(2) We provide a theoretical proof of the convergence guarantee for DivShare (Sec. 4). To the best of our knowledge, we are the first to present a formal convergence analysis in DL that captures the effect of asynchronous communication with delays. In particular, the convergence rate is influenced by the number of participating nodes, the properties of the local objective functions and initialization conditions, the parameter-wise communication rates, and the communication delays in the network.

(3) We implement and evaluate DivShare on two standard learning tasks (image classification with CIFAR-10 [34] and recommendation with MovieLens [21]) and against two state-of-the-art asynchronous DL baselines: AD-PSGD and Swift (Sec. 5). We demonstrate that DivShare is much more resilient to the presence of communication stragglers compared to the competitors and show that DivShare, with communication stragglers, speeds up the time to reach a target accuracy by up to 3.9× compared to AD-PSGD on the CIFAR-10 dataset. Compared to both baselines, DivShare also achieves up to 19.4% better accuracy and 9.5% lower test loss on the CIFAR-10 and MovieLens datasets, respectively.

## 2 Background and preliminaries

This work focuses on a scenario where multiple nodes collaboratively train ML models. This approach is often referred to as collaborative machine learning (CML) [48, 52]. In CML algorithms, each node maintains a local model and a private dataset. The private dataset is used to compute model updates and remains on the node's device throughout the entire training process.

### 2.1 Synchronous and asynchronous DL

Decentralized learning (DL) [4, 38, 44] is a type of CML algorithm in which nodes exchange model updates directly with other nodes. The majority of DL algorithms are synchronous, meaning they rely on global rounds where nodes perform computations in parallel, followed by communication with neighbors. In each round, nodes generally train their model using local data, exchange them with neighbors, and aggregate them before starting the next round. This synchronized process ensures consistency and predictable model convergence since the system progresses in synchronized rounds. However, this process also introduces inefficiencies in the presence of slower nodes (stragglers) as the system needs to wait for the slowest node to complete its computation and communication before progressing to the next round [17]. Thus, synchronous DL approaches can suffer from significant delays, particularly in settings with high variability in computing or communication speeds.

In contrast, asynchronous DL algorithms forego the notion of synchronized rounds allowing nodes to make progress independently [3]. Thus, faster nodes can continue updating and sending their models independently, which helps to mitigate the delays caused by stragglers. Designing asynchronous DL algorithms is a recent and emerging area of research [5, 9, 17, 41]. However, this asynchrony introduces new challenges. For instance, slower nodes

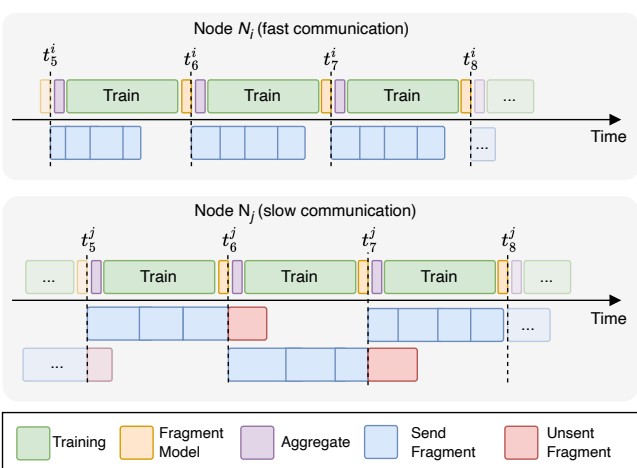

**Figure 3: Timeline of computation and communication operations in DivShare during three local rounds, from the perspective of a node with fast (top) and slow (bottom) communication. Fragments are the same number of bytes, but their transfer times may vary based on the recipient.**

spread possibly outdated model updates and slow down convergence for the entire network. The performance of local models can also get biased towards the faster nodes as their parameters are shared and mixed faster than those of the slower nodes.

### 2.2 System model

Our study specifically tackles the issue of communication delays in DL. We assume a setting where geo-distributed nodes communicate their locally trained models at their own pace independent of each other. We design DivShare to be deployed within a permissioned network setting where node participation is static. This assumption is consistent with the observation that DL is commonly used in enterprise settings, where participation is often controlled [7, 8, 12]. Nodes also remain online throughout the training process. Furthermore, we assume that nodes in DivShare faithfully execute the algorithm and consider threats such as privacy or poisoning attacks beyond scope. For clarity and presentation, we assume the nodes have comparable computation infrastructure that allows them to compute (*e.g.*, perform their local training and model aggregation) at the same speed. We further discuss this aspect in Appendix E.

### 3 Design of DivShare

We first describe the high-level operations of DivShare in Sec. 3.1 and then provide a detailed algorithm description in the remaining subsections. A summary of notations is provided in Appendix A.

### 3.1 DivShare in a nutshell

The main idea of DivShare is that nodes fragment their model and send these fragments to a diverse, random set of other nodes. Sharing models at a finer granularity allows communication stragglers to contribute at least some of the model parameters quickly while still allowing nodes with fast communication to disseminate all their model fragments. Fragmentation is also motivated by our

observations that, for an equal amount of communication, sending smaller model parts to *more* nodes results in quicker convergence than sending full models to a few nodes (see Sec. 5.3). We illustrate the workflow of DivShare in Fig. 3 where we show a timeline of operations for two nodes: $N_i$ with fast and $N_j$ with slower communication speeds. The computation and communication operations are shown in the top and bottom rows for each node, respectively, where the $k$th local round of node $N_i$ is denoted by $t_k^i$.

During a local round, each node maintains a *receive buffer* for received model parameters and a *send queue* for model fragments paired with destination identifiers, awaiting transmission. In each round, independently of others, a node $N_i$: *(i)* aggregates the model fragments received in local round $t_{k-1}^i$ (purple in Fig. 3); *(ii)* carries out SGD steps for local training (green); *(iii)* fragments the updated model to share in the next round (yellow); and *(iv)* clears the send queue and adds new pairs of model fragments and receiver identifiers. During steps *(ii)* and *(iii)*, $N_i$ continues communicating fragments of its model from local round $t_{k-1}^i$ at its own pace.

Each blue block in Fig. 3 indicates the transfer of a fragment to another node. We note that node $N_j$ with slow communication speeds may only manage to send a few fragments before it computes freshly updated model fragments, which then causes a flush of the send queue. This scenario is illustrated in Fig. 3, where unsent fragments are shown in red.

### 3.2 Problem formulation

**Decentralized learning.** We consider a set of $n \geq 2$ nodes $\mathcal{N} = \{N_1, \ldots, N_n\}$, where $N_i$ denotes the $i$th node in the network for every $i \in [n]$, who participate in this collaborative framework to train their models. $\mathcal{Z}$ denoting the space of all data points, for each $i \in [n]$, let $Z_i \subset \mathcal{Z}$, with $|Z_i| < \infty$, be the local dataset of $N_i$. Let $N_i$'s data be sampled from a distribution $\mathcal{D}^{(i)}$ over $\mathcal{Z}$ and this may differ from the data distributions of other nodes (*i.e.*, for each $z \in Z_i, z \sim \mathcal{D}^{(i)}$). Staying consistent with the standard DL algorithms, we allow each node to have their local loss functions that they wish to optimize with their personal data. In particular, setting $d \in \mathbb{N}$ as the size of the parameter space of the models, for every $i \in [n]$, let $f^{(i)} : \mathbb{R}^d \times \mathcal{Z} \mapsto \mathbb{R}_{\geq 0}$ be the loss function of node $N_i$ and it aims to train a model $x$ that minimizes $f^{(i)}(x) = \mathbb{E}_{z \sim \mathcal{D}^{(i)}} \left[ f^{(i)}(x, z) \right]$. In practice, in every local round $k$, for its local training, each node independently samples a subset, referred to as a *mini-batch*, of points from their personal dataset and seeks to minimize the average loss for that mini-batch by doing SGD steps with a learning rate $\eta > 0$. Hence, for any $i \in [n]$, letting $\xi_i^{(k)}$ to be the mini-batch sampled by $N_i$ in its local round $t_k^i$, we abuse the notation of $N_i$'s local loss for a single data point to denote the average loss for the entire mini-batch $\xi_i^{(k)}$ computed for any model $x \in \mathbb{R}^d$ given by $f^{(i)}(x, \xi_i^{(k)}) = \frac{1}{|\xi_i^{(k)}|} \sum_{z \in \xi_i^{(k)}} f^{(i)}(x, z)$. The training objective of DivShare is to collaboratively train the optimal model $x^* \in \mathbb{R}^d$ that minimizes the *global average loss*:

$$x^* = \arg\min_{x \in \mathbb{R}^d} F(x), \text{ where } F(x) = \sum_{i \in [n]} f^{(i)}(x).$$

**Asynchronous framework.** For any given $i \in [n]$ and $k = 1, 2, \ldots$, while *local rounds* $t_0^i < t_1^i, \ldots$ capture each node's progress

---

**Algorithm 1:** Local rounds in DIVSHARE from the perspective of node $N_i$

---

**1** Initialize $x^{(i,0)}$

**2** for $k = 1, \ldots, \tau$ do

**3**     $x^{(i,k)} \leftarrow$ **aggregate** $x^{(i,k-1)}$ and parameters in *InQueue*

**4**     $InQueue[j] \leftarrow \emptyset$ **for** $j$ **in** $inQueue$.keys()

**5**     $\xi_i^{(k)} \leftarrow$ mini-batch sampled from $Z_i$

**6**     $\tilde{x}^{(i,k,0)} = x^{(i,k)}$

**7**     for $h = 1, \ldots, H$ do

**8**        $\tilde{x}^{(i,k,h)} \leftarrow \tilde{x}^{(i,k,h-1)} - \eta \nabla f^{(i)}\left(\tilde{x}^{(i,k,h-1)}, \xi_i^{(k)}\right)$

**9**     FRAGMENTMODEL$\left(\tilde{x}^{(i,k,H)}\right)$ // See Alg. 2

**10** return $x^{(i,\tau)}$

---

**Algorithm 2:** Model fragmentation in DIVSHARE from the perspective of node $N_i$

---

**Require:** Fragmentation fraction $\Omega$, nodes $\mathcal{N}$, number of fragment recipients $J$, sending queue *OutQueue*

**1** **Procedure** FRAGMENTMODEL($x$):

**2**     $OutQueue \leftarrow \emptyset$

**3**     **Fragment** $x$ into $\left\lceil \frac{1}{\Omega} \right\rceil$ fragments

**4**     for *each fragment $f$* do

**5**        $S \leftarrow$ **Sample** $J$ random nodes from $\mathcal{N}$

**6**        for *each sampled node $N_j$ in $S$* do

**7**           $OutQueue$.add($(j, f)$)

**8**     SHUFFLE($OutQueue$)

---

in its compute operations and communication (as described in Sec. 3.1), we introduce the notion of *global rounds* to track the network's overall training progress. Similar to the formalization made by Even et al. [17], we define global rounds $T_0 < T_1 \ldots$, where at each $T_k$, a non-empty subset of nodes update their models (*i.e.*, aggregate received models, perform local SGD steps, and fragment the updated model). Between two global rounds, $T_k$ and $T_{k+1}$, none to plenty of communication may asynchronously occur between nodes. We assume that communication times between any pair of nodes are independent and directional. That is, for any two nodes $N_i, N_j \in \mathcal{N}$, the time it takes for $N_i$ to communicate with $N_j$ may differ from the time it takes for $N_j$ to communicate with $N_i$.

In order to lay down the algorithmic workflow of DIVSHARE, for every $i \in [n]$, let the model held by node $N_i$ in global round $T_k$ for $k \in \{0, 1, \ldots\}$ be denoted by $x^{(i,k)} \in \mathbb{R}^d$ with $x_\iota^{(i,k)}$ being the $\iota$th parameter of $x^{(i,k)}$ for each $\iota \in [d]$.

### 3.3 The DIVSHARE algorithm

We now formally describe the DIVSHARE algorithm from the perspective of node $N_i$. A node in DIVSHARE executes three processes in parallel and independently of other nodes: (*i*) model aggregation, training and fragmentation (computation tasks, see Alg. 1 and Alg. 2), (*ii*) model receiving (Alg. 3), and (*iii*) model sending (Alg. 3). As mentioned before, each node keeps track of a receive buffer with incoming model fragments it has received during a local round,

referred to as *InQueue*, and a send queue with model fragment and node destination pairs, referred to as *OutQueue*.

**Computation tasks.** We formalize the computation tasks that a node $N_i \in \mathcal{N}$ conducts in Alg. 1. $N_i$ first initializes model $x^{(i,0)}$, and all nodes independently do this model initialization. In each of the $k = 1, \ldots, T$ local rounds, where $T$ is a system parameter, $N_i$ first performs parameter-wise aggregation of all parameters in $x^{(i,k-1)}$ and all received fragments present in *InQueue* that were received in the previous round with uniform weights (Line 3). We note that the count of each received parameter by $N_i$ may differ. $N_i$ then resets *InQueue* (Line 4) and starts updating its model by performing $H$ local SGD steps using a mini-batch sampled from its local dataset. The resulting model $x^{(i,k)}$ is then fragmented (Line 9), and the fragments are shared with other nodes in parallel to the computation during the next local round (as shown in Fig. 3).

Alg. 2 outlines how nodes in DIVSHARE fragment their models and fill the send queue. Model fragmentation is dictated by the fragmentation fraction $\Omega$, which specifies the granularity at which a model is fragmented. Specifically, a model is fragmented in $\left\lceil \frac{1}{\Omega} \right\rceil$ fragments. This resembles random sparsification, a technique used in CML to reduce communication costs [8, 23, 31, 53]. For each fragment $f$, we uniformly randomly sample $J$ other nodes to send this fragment to (Line 5), and add each recipient node $N_j$ and the corresponding fragment as tuple $(j, f)$ to sending queue. Finally, we shuffle the order of *(destination, fragment)* pairs in the sending queue. This shuffling is important to ensure that slow nodes send diverse sets of model parameters within a single, local round. While we uniformly random shuffle the sending queue in our experiments, we acknowledge that different shuffling strategies can be used, *e.g.*, we could prioritize the sending of more important parameters as done in sparsification [2, 15, 30].

**Communication tasks.** For every $i, j \in [n]$ with $i \neq j$, when receiving a model fragment $f$ from $N_j$, $N_i$ adds the parameters in $f$ to the receive buffer *InQueue*, associated with node $N_j$. If a parameter $\iota$ is received twice from $N_j$ during a particular local round, the older parameter $InQueue[j][\iota]$ will be replaced with the latest one. In addition, $N_i$ runs a sending loop where it continuously sends information in *OutQueue*. Specifically, $N_i$ pops the tuple $(j, f)$ from *OutQueue* and sends fragment $f$ to node $N_j$. Due to space constraints, we provide the associated logic in Alg. 3 in Appendix D.

## 4 Convergence analysis

In this section, we theoretically analyze the convergence guarantees of DIVSHARE from the perspective of the global rounds. As discussed in previous works on asynchronous decentralized optimization [17, 32], in order to allow the nodes to hold heterogeneous data and their individual local objective functions, we need to make assumptions on the computation sampling. In particular, recalling that this work focuses on communication straggling in the network, we assume computation homogeneity *i.e.*, every node computes in each round.

**Notations.** $\|\cdot\|$ denotes the Euclidean norm for a vector and the spectral norm for a matrix. A function $f$ is *convex* if for each $x, y$ and subgradient $g \in \partial f(x)$, $f(y) \geq f(x) + \langle g, y - x \rangle$. When $f$ and $f^{(i)}$ are convex, we do not necessarily assume they are differentiable, but we abuse notation and use $\nabla f(x, \xi)$ and $\nabla f^{(i)}(x)$ to denote an arbitrary subgradient at $x$. $f$ is *B-Lipschitz-continuous* if for any

$x, y \in \mathbb{R}^d$ and $z \in \mathcal{Z}$, $|f(x, z) - f(y, z)| \le B\|x - y\|$. $f$ is $L$-*smooth* if it is differentiable and its gradient is $L$-*Lipschitz-continuous*.

We assume that in each round every node independently connects with other nodes to share each of its model fragments. In particular, for any $i, j \in [n]$, let the probability that $N_j$ shares each of its model fragments with $N_i$ be $\frac{J}{n-1}$, making $J$ the expected number of nodes that receive each of $N_j$'s model parameters.

To capture the effect of communication stragglers, let $k_{ji}$ denote the number of global rounds it takes for node $N_j$ to send one of its model fragments to node $N_i$. Consequently, define $K_j = \max_{1 \le i \le n} k_{ji}$ as the maximum delay for node $N_j$ to communicate with its neighbors, $K = \max_{1 \le j \le n} K_j$ as the global maximum communication delay, and $T = \sum_{1 \le j \le n} K_j$ as the total communication delay. We assume $K$ (and, therefore, $T$) to be finite. For any $\iota \in [d]$, the model update $x_\iota^{(i,k)}$ is aggregated as:

$$x_\iota^{(i,k)} = \frac{1}{1 + R_\iota^{i,k}} \sum_{1 \le j \le n} x_\iota^{(j,k-k_{ji})} \mathbf{1}\left(A_\iota^{j,k-k_{ji},i}\right) \tag{1}$$

where $R_\iota^{i,k} = \sum_{1 \le j \le n, j \ne i} \mathbf{1}\left(A_\iota^{j,k-k_{ji},i}\right)$ with $A_\iota^{j,k-k_{ji},i}$ being the event where $N_j$ shared its model parameter $\iota$ in a fragment with $N_i$ in round $k - k_{ji}$. Note that $1 + R_\iota^{i,k}$ is the normalization factor and is always greater than 1 as the buffer always contains the $N_i$'s own model.

We refer to $X_\iota^k = \left(x_\iota^{(i,k-k_i)}\right)_{\substack{1 \le i \le n, \\ 1 \le k_i \le K_i}}$ as the *sliding window* of the network-wide $\iota$th model parameter containing all the information needed to generate a new global step in the algorithm, we enable ourselves to write losses on the sliding window as the vector of the losses. This communication step can be encoded by a random matrix $W_\iota^k$ representing the shift of the sliding window from $\left(x_\iota^{(i,k-k_i)}\right)_{\substack{1 \le i \le n, \\ 1 \le k_i \le K_i}}$ to $\left(x_\iota^{(i,k+1-k_i)}\right)_{\substack{1 \le i \le n, \\ 1 \le k_i \le K_i}}$, by generating a new step using Eq. (1). Setting $\alpha_\iota^{j,k-k_{ji},i} = \frac{\mathbf{1}\left(A_\iota^{j,k-k_{ji},i}\right)}{1+R_\iota^{i,k}}$ as the initial weight, for every $i, j \in [n]$ and $1 \le k_j \le K_j$, formally $W_\iota^k$ can be expressed as:

$$\left(W_\iota^k\right)_{(i,k_i),(j,k_j)} = \begin{cases} \delta_{i,j}\delta_{k_i-1,k_j} \frac{1}{1+R_\iota^{i,k}} & \text{if } 2 \le k_i \le K_i \\ \delta_{k_j,k_{ji}} \alpha_\iota^{j,k-k_{ji},i} & \text{when } k_i = 1 \end{cases} \tag{2}$$

where, for any $\alpha, \beta \in \mathbb{R}$, $\delta_{\alpha,\beta}$ is equal to 1 when $\alpha = \beta$, or 0 otherwise. For any $k \ge K$ and $\tilde{k} \ge 1$, let $W_\iota^{(k:k+\tilde{k}-1)} = \left(W_\iota^{k+\tilde{k}-1} \dots W_\iota^k\right)$.

To develop the theoretical study of convergence of DivShare, in addition to assuming the existence of the minimum of the loss function $F$, we make the standard set of assumptions (Assumptions 1 to 3) that are widespread in related works [8, 16, 17] and introduce Assumption 4 that is specific to the environment with asynchronous and delayed communication that we are considering.

*Assumption* 1 (Condition on the objective). Denoting the minimum loss over the entire model space as $F^* = \min_{x \in \mathbb{R}^d} F(x)$, let $\Delta$ be an upper bound on the *initial suboptimality*, i.e., $\Delta \ge \left\|F\left(X^{(0)}\right) - F^*\mathbf{1}\right\|$, and let $D$ be an upper bound on the *initial distance* to the minimizer, i.e., $D \ge \min\left\|X^{(K)} - (x^*)\mathbf{1}\right\|$, where $x^* = \arg\min_{x \in \mathbb{R}^d} F(x)$ is the model that minimizes the loss and is assumed to exist.

*Assumption* 2 (Condition on gradients). There exists $\sigma^2 > 0$ such that for all $x \in \mathbb{R}^d$, $i \in [n]$, we have $\mathbb{E}_{\xi \sim \mathcal{D}^{(i)}}\left[\nabla f^{(i)}(x, \xi)\right] = \nabla f^{(i)}(x)$ and $\mathbb{E}_{\xi \sim \mathcal{D}^{(i)}}\left[\left\|\nabla f^{(i)}(x, \xi) - \nabla f^{(i)}(x)\right\|^2\right] \le \sigma^2$.

*Assumption* 3 (Heterogeneous setting). There exists $\zeta^2 > 0$ such that the *population variance* is bounded above by $\zeta^2$, i.e., for all $x \in \mathbb{R}^d$, we have $\sum_{i \in [n]} \left\|\nabla f^{(i)}(x) - \nabla F(x)\right\|^2 \le \zeta^2$.

*Assumption* 4 (Straggling-communication balance). For any $i \in [n]$ and $k \in \mathbb{Z}_{\ge 0}$, let $\alpha_{(1)} = \mathbb{E}\left[\frac{1}{1+R^{i,k}}\right] = \frac{n-1}{Jn}\left(1 - \left(1 - \frac{J}{n-1}\right)^n\right)$ and $\alpha = \frac{1}{n-1}\left(1 - \alpha_{(1)}\right)$. Then we assume that $(T - n)\left(\frac{(\alpha n)^2}{T} + \alpha_{(1)}^2\right) < 1$.

*Remark* 1. Observing that $T - n$ equals 0 when the system is synchronous and increases as the sum of the delays grows, it effectively parameterizes the total amount of *straggling* in the system. On the other hand, the term $\left(\frac{(\alpha n)^2}{T} + \alpha_{(1)}^2\right)$ can be interpreted as the *communication rate* – it decreases as $J$, the expected number of neighbors to which a node sends each of its model parameters, increases. This implies that communication speeds up as nodes engage in more frequent interactions within the network. Assumption 4 essentially strikes a balance by bounding the combined effects of straggling and the communication rate in the network. Appendix G discusses the asymptotic properties of Assumption 4 and provides analytical insights into the relationship between the average communication delays and the number of nodes in the network. This, in turn, demonstrates the practicality of adopting Assumption 4 in real-world settings and highlights the robustness of DivShare in handling communication stragglers.

*Theorem* 1 (Convergence of DivShare). Under Assumptions 1 to 4 and if $f^{(i)}$ is $L$-smooth for all $i \in [n]$, then $\mathbb{E}\left[\frac{1}{\tilde{k}} \sum_{k < \tilde{k}} \left\|\nabla F\left(\overline{X^k}\right)\right\|^2\right]$

$$= O\left(\left(\frac{\hat{L}\left(\sigma^2 + \zeta^2\right)}{\tilde{k}}\right)^{\frac{1}{2}} + \left(\frac{n\hat{L}\sqrt{\sigma^2\Lambda + \zeta^2\Lambda^2}}{\tilde{k}}\right)^{\frac{2}{3}} + \frac{\hat{L}\left(n^{-\frac{1}{2}} + \Lambda\right)}{n\tilde{k}}\right),$$

where $\lambda_2 = \|\mathbb{E}[W]\Pi_F\|$ with $\Pi_F$ being the canonical projector on $F = \mathbf{1}^\perp$, $\Lambda = (\alpha|\log(\lambda_2)| + (1 - \alpha)\log(T))\alpha^{-1}|\log(\lambda_2)|^{-2}$, and $\hat{L} = L\Delta$. In the interest of space, the proof is postponed to Appendix F.

*Remark* 2. Th. 1 essentially shows how fast the average model of all the nodes parameter-wise converges. First, the slowest term, number-of-steps wise, does not depend on $\Lambda$ thus on the delays, achieving the sub-linear rate of $O\left(1/\sqrt{\tilde{k}}\right)$ rounds, optimal for SGD, we achieve this by considering the convergence over a time sliding-window. For the numerators of the second and third terms, the rate is encoded in Assumption 4 with the coefficient $\Lambda$ going to zero as $(T - n)\left(\frac{(\alpha n)^2}{T} + \alpha_{(1)}^2\right)$ goes to 0 (see Eq. (4) in Appendix F).

## 5 Evaluation

We now describe our experimental setup (Sec. 5.1), compare DivShare to baselines (Sec. 5.2), evaluate the sensitivity of DivShare to its parameters (Sec. 5.3), and quantify the performance of DivShare and baselines in real-world network conditions (Sec. 5.4).

## 5.1 Experimental setup

**Implementation.** We implement DivShare in Python 3.8 over DecentralizePy [14] and PyTorch 2.1.1 [49] to emulate DL nodes. For emulating communication stragglers, we used the Kollaps [19] network simulator to control the latency and bandwidth of each network link. The source code will be made available on GitHub.

**Network setup.** We conduct experiments with 60 nodes that can communicate with all other nodes. Unless otherwise stated, we group nodes into *fast* and *straggler* nodes. We set a fixed bandwidth and latency (1 ms) for all *fast* nodes. The mean bandwidth of fast nodes is set at 60 MiB/s and 200 MiB/s for all the experiments involving the CIFAR-10 and MovieLens dataset, respectively. To systematically study the effect of varying degrees of straggling, we introduce a (communication) straggling factor $f_s$. The bandwidth of the *straggler* nodes is sampled from a normal distribution with a mean $f_s$ times lower than the bandwidth of *fast* nodes and a standard deviation of 0.5.

**Baselines.** We compare DivShare to two state-of-the-art asynchronous DL algorithms: Swift [9] and AD-PSGD [39]. AD-PSGD is a standard asynchronous DL algorithm where nodes, similar to DivShare, independently progress in local rounds. In each local round of AD-PSGD, a node $N_i$ updates its model and selects a single neighbor $N_j$, after which $N_i$ and $N_j$ bilaterally average their local models. Swift also allows nodes to work at their own speed. However, unlike AD-PSGD, Swift operates in a wait-free manner, *i.e.*, nodes do not wait for simultaneous averaging among nodes. Instead, a node asynchronously aggregates multiple models it has received from its neighbors, eliminating the need for synchronization. Moreover, Swift uses a non-symmetric and non-doubly stochastic communication matrix, updated dynamically during training.

For DivShare, we use a fragmentation fraction $\Omega = 0.1$, unless specified otherwise. Thus, each node splits its model into 10 equally-sized fragments before sending these to other nodes (see Alg. 2). We set a degree of $J = \lceil \log_2(n) \rceil = 6$ for DivShare, a common choice in random topologies [12], *i.e.*, each node sends each fragment to 6 other nodes every local round. We use the same model exchange characteristics for Swift, *i.e.*, each node sends its full model to 6 other nodes every local round.

**Task.** We evaluate DivShare and baselines over two common learning tasks: image classification and recommendation. For the former, we use the CIFAR-10 [34] dataset and a GN-LeNet model [24, 37]. We introduce label heterogeneity by splitting the dataset into small shards and assigning each node a uniform share of the shards [14, 15, 42]. This partitioning ensures that each node receives the same number of training samples, but the number of shards allows us to control the heterogeneity: the higher the number of shards, the more uniform the label distribution becomes. Unless stated otherwise, we assign 5 shards to each node. For the recommendation task, we use the MovieLens 100K [21] dataset and a matrix factorization model [33]. For CIFAR-10, we report the average top-1 test accuracy, while for MovieLens, we report the MSE loss between the actual and predicted ratings. We run each experiment 3 times with different seeds and present the averaged results. We provide additional experiment details in Appendix B.

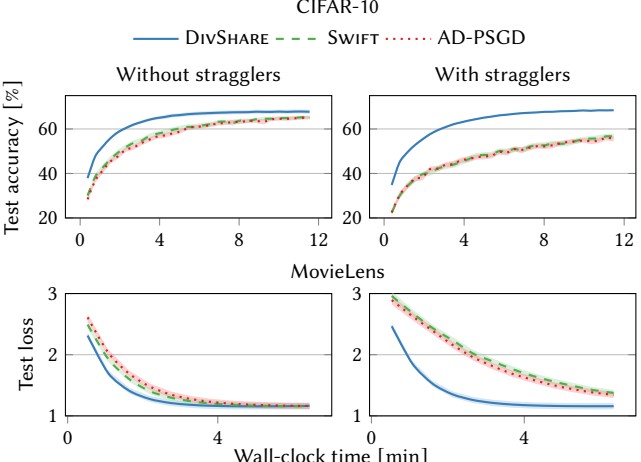

**Figure 4: The model utility of DivShare and the baselines over time, with and without stragglers on CIFAR-10 (↑ is better) and MovieLens (↓ is better).**

## 5.2 Convergence of DivShare against baselines

We first evaluate the convergence of DivShare against the baselines. Fig. 4 shows the evolution of model utility over time, for both CIFAR-10 and MovieLens, in a setting without stragglers (with $f_s = 1$) and where half of the nodes are stragglers (with $f_s = 5$). Both baselines show comparable performance in all settings. We also observe that communication stragglers significantly slow the convergence of both AD-PSGD and Swift. Specifically, in the presence of communication stragglers, the baselines reach 12.5% and 10.2% worse model utilities in CIFAR-10 and MovieLens, respectively, compared to the scenario without stragglers. DivShare outperforms both baselines in terms of the speed of convergence and model test utility across both datasets. The superior performance of DivShare is especially evident in the presence of stragglers, achieving up to 19.4% relatively better accuracy and 9.5% lower test loss for the CIFAR-10 and MovieLens datasets, respectively. We attribute this to the ability of DivShare to effectively aggregate models in small fragments and use the available bandwidth effectively.

## 5.3 Sensitivity analysis

In the following, we analyze the effect of the straggling factor $f_s$, varying levels of non-IIDness, and the fragmentation fraction $\Omega$ on the performance of DivShare and baselines.

**Varying the degree of communication straggling.** We next explore the effect of the straggling factor $f_s$ and varying number of stragglers on the performance of AD-PSGD and DivShare. Fig. 5 shows the heatmaps for the *(a)* final test accuracy after 15 min and *(b)* wall-clock time to achieve 60% test accuracy on CIFAR-10 for a varying number of stragglers and increasing $f_s$. Similar to Fig. 4, we observe that communication stragglers hinder the convergence of AD-PSGD. With only $n/8$ (7 of the 60) nodes being communication stragglers, increasing $f_s$ from 1 to 5 increases the time to reach the target accuracy by 43.4%. AD-PSGD is unable to attain the target accuracy of 60% with $n/2$ (30 out of 60) communication

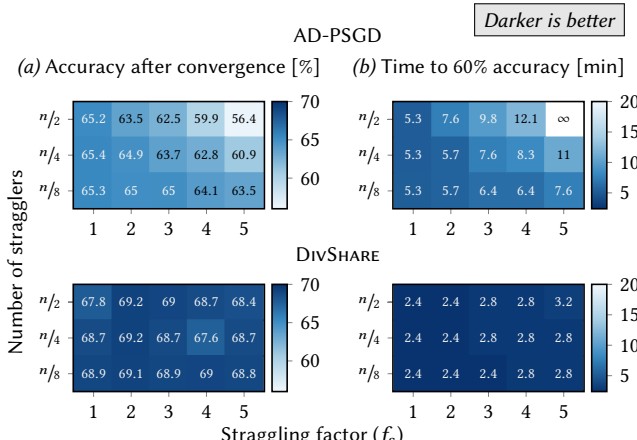

**Figure 5: (a) Accuracy after convergence and (b) time to** 60% **accuracy on CIFAR-10. A time to accuracy of** $\infty$ **means that AD-PSGD did not reach the target accuracy.**

stragglers. In contrast, DivShare displays minimal deviation from an ideal setting without stragglers as the number of stragglers and $f_s$ increases. As shown in Fig. 5(a), DivShare consistently achieves better test accuracy compared to AD-PSGD and shows a speedup of at least 2.2× over AD-PSGD. In the case of 15 stragglers ($n/4$) and $f_s$ = 5, DivShare achieves a speedup of 3.9× over AD-PSGD to reach the same accuracy. Experiments with the MovieLens dataset shows similar trends and are provided in Appendix C. To conclude, DivShare retains its strong performance even when half of the nodes are up to 5× slower in communication than the others.

**Effect of data heterogeneity.** We analyze the effect of varying levels of data heterogeneity and $f_s$ on the time-to-accuracy speedup in DivShare. Fig. 6(a) shows the heatmap of the speedup due to fragmentation, *i.e.*, the percentage improvement in the time to reach 60% test accuracy for $f_s$ = 0.1 vs. $f_s$ = 1 with 30 stragglers. Each row represents decreasing data heterogeneity, with 10 being almost IID. Fig. 6(a) reveals that fragmentation in DivShare is advantageous at all data heterogeneity levels and straggling factors. However, the speedup owing to fragmentation is amplified at high heterogeneity levels and high levels of straggling, achieving a speedup of up to 84%. In other words, as the learning task gets more difficult, DivShare shows more efficiency and resilience to stragglers.

**Limits of fragmentation.** The granularity of fragmentation, controlled by $\Omega$, is an important parameter in DivShare. To understand the limits of fragmentation in DivShare, we evaluate DivShare with different values of $\Omega$, ranging from 0.01 to 1, 30 stragglers, and $f_s$ = 5. Fig. 6(b and c) shows the time required to reach a target accuracy of 60% on CIFAR-10 without and with stragglers. In both cases, as we decrease the $\Omega$ from 1, *i.e.*, as nodes start sending smaller fragments, the convergence speed improves until $\Omega$ = 0.1. With a low $\Omega$ value, each node potentially sends a fragment to all other nodes. While further decreasing $\Omega$ does not alter the dissemination of information, it does increase the number of messages flowing in the system. Therefore, at fragmentation factors < 0.1, we see a rapid increase in the time to convergence due to

the large number of messages leading to network congestion and overhead in the case without stragglers. The effect is ameliorated in the scenario with stragglers (Fig. 6(c)) since the straggling nodes rarely send out all the fragments in a round.

**Varying the straggling factor and fragmentation fraction.** We assess the effects of varying $f_s$ on the attained model utility and the time until 60% accuracy is reached on CIFAR-10, for baselines and DivShare with varying fragmentation fractions. Fig. 6(d and e) show the final accuracy and the wall-clock time to 60% test accuracy, respectively, on CIFAR-10 for DivShare and the baselines. Intuitively, the attained final accuracy and the convergence rate deteriorate for the baselines AD-PSGD and Swift as $f_s$ increases. We observe the same trend for DivShare at higher fragmentation factors (resulting in less granular fragments). DivShare with a fragmentation factor of 0.1, however, demonstrates strong robustness to stragglers, achieving almost the same accuracy across the spectrum at a small increase in the time to target accuracy.

In summary, we observe in Fig. 6(a-d) a sweet spot for $\Omega$ in DivShare around $J/n$, corresponding to 0.1 in our experiment setup. For this value, DivShare exhibits the highest model utility and convergence speeds on the CIFAR-10 dataset.

## 5.4 Real-world network evaluation

So far, we have evaluated DivShare and baselines by emulating communication stragglers using the straggling factor $f_s$. We next evaluate DivShare in a real-world scenario using realistic network characteristics reported in the work of Gramoli et al. [20]. This work provides a bandwidth and latency matrix between each pair of 10 AWS regions [20]. We integrate these matrices in our experiment setup and place 6 random nodes in each region. Fig. 7 shows the model convergence for CIFAR-10 (left) and MovieLens (right) for DivShare and baselines. This figure shows that DivShare outperforms AD-PSGD and Swift on both datasets in terms of convergence time. This is particularly evident on the CIFAR-10 dataset, where DivShare reaches 60% accuracy, 35.6% faster than baselines. Fig. 7 thus demonstrates that DivShare outperforms the baselines in real-world networks as well.

## 6 Related work

**Synchronous DL.** Most DL algorithms are synchronous and tend to perform sub-optimally when faced with variance in computing and communication speed of nodes. Decentralized parallel stochastic gradient descent (D-PSGD) [38] remains a widely used synchronous DL baseline, where nodes perform model updates in lockstep. In order to improve model convergence time, numerous approaches optimize the communication topology before training begins [36, 51, 57] or dynamically throughout training [12]. DivShare employs a randomized communication strategy instead of a fixed topology, allowing nodes to share model fragments with potentially any other node.

**Asynchronous DL.** Asynchronous DL approaches have gained attention as they improve resource utilization. Early works in asynchronous optimization focus on improving convergence rates in distributed settings [3, 46, 56]. Asynchronous methods have more recently also been applied to DL. Solutions like AD-PSGD [39],

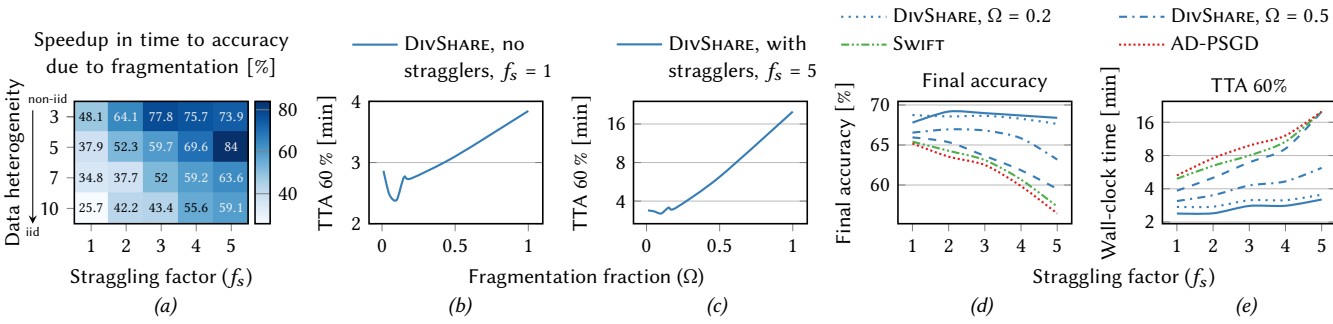

Figure 6: (a) Speedup in DivShare due to fragmentation at different levels of data heterogeneity. Darker is better. (b) and (c) Impact of the fragment fraction $\Omega$ on time to accuracy (TTA) with and without stragglers (lowest TTA is achieved around $\Omega = 0.1$). (d) and (e) Impact of the straggling factor on the final accuracy and time to target accuracy for DivShare and baselines.

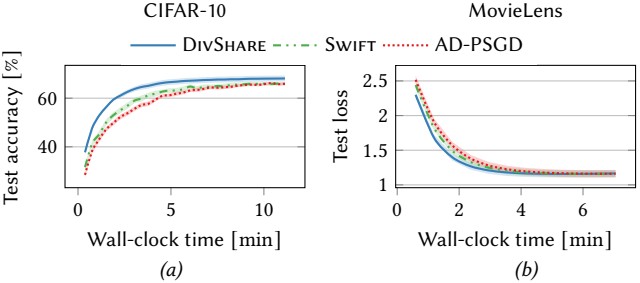

Figure 7: The model utility of DivShare and baselines over time, under real-world network conditions [20].

SwarmSGD [43] and Swift [9] have tackled this problem by allowing nodes to perform updates independently, thereby avoiding idle time and speeding up convergence. Gossip learning (GL) is another DL approach where nodes periodically update their model, send it to another random node in the network, and use a staleness-aware aggregation method to merge model updates [22, 47]. Asynchronous DL has also been explored in wireless and edge computing where system heterogeneity is common [28, 29]. Many of the works mentioned above assume idealized communication scenarios [56], which can lead to sub-optimal performance under real-world conditions such as network latency and bandwidth. In contrast, our work is targeted to real-world geo-distributed networks.

Asynchrony is also a popular research topic in federated learning (FL) [55]. Asynchronous FL systems such as FedBuff [45], Payapa [26], Fleet [11], and REFL [1] use staleness-aware parameter aggregation, ensuring that updates from slower devices do not adversely affect model performance. In these systems, aggregation is performed by a parameter server, whereas DivShare is decentralized and leverages peer-to-peer communication.

**Sparsification.** Model fragmentation in DivShare is conceptually similar to sparsification techniques [2, 50]. These techniques reduce the communication load by sharing only a subset of model parameters. In DivShare, nodes share a varying number of model parameters in a single local round, depending on their computing

and communication speed. This differs from sparsification, where typically a fixed portion of parameters is sent per round. Model fragmentation has also been used to improve privacy in DL [8]. DivShare instead uses fragmentation to optimize model convergence and add resilience to communication stragglers.

## 7 Final remarks

We presented DivShare, a novel and asynchronous DL algorithm that improves performance and convergence speed, especially in networks having communication stragglers. The key idea is that each node fragments its model and sends each fragment to a random set of nodes instead of sending the full model. We theoretically proved the convergence of DivShare using a novel technique, making it the first formal convergence analysis in DL to incorporate asynchronous communication with stragglers. Finally, we empirically demonstrated with two learning tasks that DivShare can achieve up to 2.2× speedup and 19.4% better accuracy in the presence of data heterogeneity and up to half of the network straggling.

**Number of messages.** Fragmentation of models in DivShare allows straggling nodes to contribute their locally trained models to the network. While DivShare has the same communication cost as other DL approaches in terms of bytes transferred, the benefits come at the cost of an increased number of messages sent by the nodes. Since the nodes transfer large models in DL training, the increased latency due to the number of messages does not negatively affect real-world systems, as shown in Sec. 5.4. More advanced fragmentation methods can be used to reduce the number of messages, which is an interesting avenue for future research.

**No synchronization barriers.** In this work, we assumed DL nodes to have similar hardware and hence, similar compute speeds. However, DivShare has no synchronization barriers. In our experiments, all nodes run DivShare with similar hardware, but the OS scheduling and jitter introduce small drifts in the speeds of the nodes. We observed that nodes can be up to a few local rounds apart in DivShare. Therefore, DivShare can handle communication stragglers without synchronization barriers and uniform computation speeds, as shown in Sec. 5. Handling nodes with vastly different computation speeds is interesting future work.

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

## A Table of notations

| Notation | Description |
|---|---|
| $\mathcal{N}$ | Set of all nodes in the network |
| $n$ | Total number of nodes (i.e., $|\mathcal{N}|$) |
| $N_i$ | Node $i$ for $i = 1, \ldots, n$ |
| $\mathcal{D}^{(i)}$ | Data distribution of $N_i$ |
| $Z_i$ | Local dataset of $N_i$ |
| $f^{(i)}$ | Local loss function of $N_i$ |
| $F$ | Global average loss |
| $\{t_0^i, t_1^i, \ldots\}$ | Local rounds of $N_i$ |
| $\xi_i^{(k)}$ | Mini-batch sampled by $N_i$ in $k$th local round |
| $\eta$ | Learning rate used for local SGD steps |
| $H$ | No. of local SGD steps performed by each node |
| $\{T_0, T_1 \ldots\}$ | Global rounds tracking progress of all nodes |
| $d$ | size of the parameter space of the models |
| $x^{(i,k)}$ | $N_i$'s model in global round $T_k$ |
| $x_\iota^{(i,k)}$ | $\iota$th parameter of $x^{(i,k)}$ for all $\iota \in [d]$ |
| $\Omega$ | Fragmentation fraction for each node |
| $J$ | No. of nodes each model fragment is shared with |
| $f_s$ | Straggling factor |
| $T$ | Total communication delay (w.r.t. global rounds) |
| $K$ | Global maximum communication delay |
| $X_\iota^{(k)}$ | Sliding window of network-wide $\iota$th model parameter used to generate global round $T_k$ |

## B Experimental details

Table 1 provides a summary of the datasets we used for our evaluation in Sec. 5, along with various hyperparameters and settings. We perform experiments on respectively 5 and 10 hyperthreading-enabled machines, for CIFAR-10 and MovieLens, respectively. Each machine is equipped with 32 dual Intel(R) Xeon(R) CPU E5-2630 v3 @ 2.40GHz cores. The network speeds of fast nodes, along with number of rounds and batch size, were tuned (i) to reach the best performances, (ii) such that in a system without stragglers, the time to send all the messages from a node is the time to perform one computation round, enabling us, when there are stragglers, to highlight the straggling effect and the delays induced in the network. Fig. 8 shows how the bandwidth of network links change in the presence of communication stragglers. The nodes can simultaneously receive messages from multiple nodes with the total bandwidth capped by the network link.

## C Additional experiments

Fig. 9 shows the heatmap for the loss after convergence and time to a target loss for DivShare and AD-PSGD on MovieLens. The experimental setup is the same as described in Appendix B. We vary the number of stragglers in the network and the degree of straggling. Similar to the results in the main text (Sec. 5.3), we observe that DivShare outperforms the baselines, and the gains become more significant as the task becomes more difficult (high straggling).

## D Communication logic of DivShare

We provide the communication logic of DivShare in Alg. 3. This logic shows the procedure called when a node $N_i$ receives a model fragment from $N_j$. We also show the sending loop, which is continuously executed by $N_i$. The sending loop obtains a fragment $f$ and the index of the recipient node $j$ from the *OutQueue* and then sends $f$ to node $N_j$.

---

**Algorithm 3:** Communication logic in DivShare from the perspective of node $N_i$

**1 Procedure** *onReceiveFragment($f$, $j$)*:
**2**    // We received fragment $f$ from node $N_j$
**3**    **for** *Parameter $\iota$* **in** $f$ **do**
**4**       $InQueue[j][\iota] \leftarrow \iota$
**5 Sending Loop**
**6**    $(j, f) \leftarrow OutQueue$.pop()
**7**    **Send** fragment $f$ to node $N_j$
**8 End Loop**

---

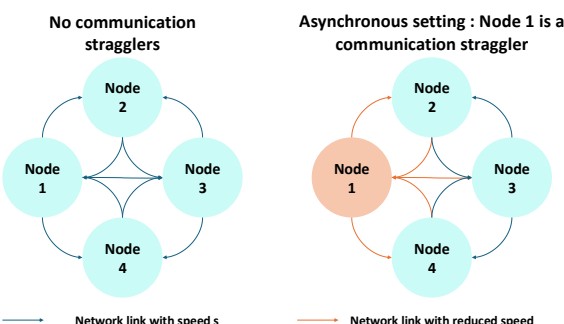

**Figure 8: Example of how the network speeds of fast and straggling nodes are changed in our experiments. In a network without stragglers (left), all network links have a bandwidth of $s$. When node 1 becomes a straggler (right), the bandwidth capacity of its network links are reduced to $\frac{s}{f_s}$ where $f_s$ is the straggling factor.**

## E Uniform compute speeds

As the focus of DivShare is on communication stragglers, we focus on a setting where nodes have uniform compute speeds. We argue that in DL environments, it is feasible to control the hardware characteristics of nodes, especially in enterprise networks where our framework is designed to operate. Nodes can purchase similar hardware or standardize on specific processing units, thus achieving roughly uniform compute speeds. Network speeds, however, are inherently more challenging to control due to factors like network congestion and geographical distances. This assumption has enabled the theoretical analysis presented in Sec. 4.

DivShare can function in heterogeneous computing environments as well. Each node $i$ can set the time between two executive

**Table 1: Summary of datasets and associated hyperparameters used in our evaluation.**

| Task | Dataset | Model | $\eta$ | b | Training Samples | Number of rounds per iteration | Number of Iterations | Fast nodes Network speed |
|------|---------|-------|--------|---|------------------|-------------------------------|---------------------|--------------------------|
| Image Classification | CIFAR-10 | GN-LeNet | 0.050 | 8 | 50 000 | 128 | 350 | 60 Mbps |
| Recommendation | MovieLens | Matrix Factorization | 0.050 | 2 | 70 000 | 400 | 650 | 200 Mbps |

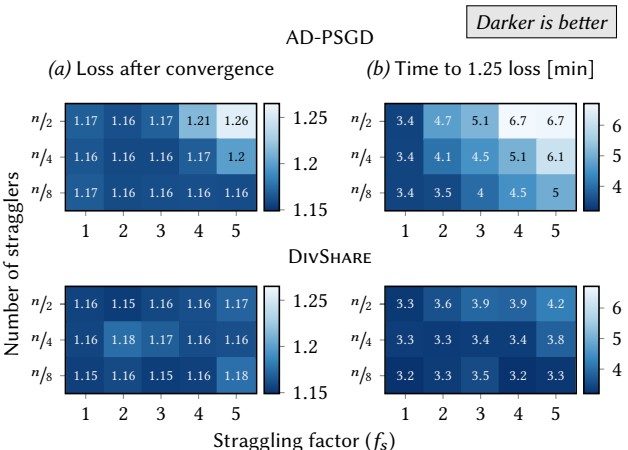

Figure 9: *(a)* Final test loss after convergence and *(b)* time to 1.25 test loss on MovieLens with $n = 60$.

local rounds $t_k^i$ and $t_{k+1}^i$ to the time that the slowest node in the network requires to update its model. Even if a node with fast computing speed finishes training before the next local round starts, it can send its model fragments to other nodes. This approach, however, can introduce idle compute or communication time. We leave a theoretical and experimental analysis in settings with both compute and communication stragglers for future work.

## F   Proof of Theorem 1

We first derive Lem. 2 that shows that the communication matrix of DivShare satisfies *Ergodic mixing* and this, in turn, acts as a crucial intermediate step leading to our main result.

*Lemma* 2 (Ergodic mixing of DivShare). If Assumptions 3 and 4 hold, we have $\lambda_2 < 1$ and, for every $\rho \in (0, 1)$, setting

$$k_\rho = \left( \frac{\sqrt{2 \log(T) \frac{1-\alpha}{\alpha}} + \sqrt{2 \log(T) \frac{1-\alpha}{\alpha} + 8 \log(\lambda_2) \log(1-\rho)}}{2|\log(\lambda_2)|} \right)^2,$$

we have, for every $\tilde{k} \geq k_\rho$, $k \geq K$, and $X \in \mathbb{R}^T$:

$$\mathbb{E}\left[ \left\| W_\iota^{(k:k+\tilde{k}-1)} X - \overline{X} \right\|^2 \right] \leq (1-\rho)^2 \left\| X - \overline{X} \right\|^2.$$

**Proof Lemma 2.** Our goal is to show that:

$$\forall \rho \in (0,1), \exists \tilde{k} \in \mathcal{N}, \forall k \geq K, \forall X \in \mathbb{R}^T$$

$$\mathbb{E}\left[ \left\| W_\iota^{(k:k+\tilde{k}-1)} X - \overline{X} \right\|^2 \right] \leq (1-\rho)^2 \left\| X - \overline{X} \right\|^2$$

Let $F = \mathbf{1}^\perp$, $\Pi_F$ the canonical projector on $F$ (respectively $\Pi_{\mathbf{1}}$ the projector on $\mathbb{R}\mathbf{1}$) we have for $X \in \mathbb{R}^T, k \geq K$

$$W^k X - \overline{X} = W^k \left( X - \overline{X} \right)$$

Let's remark that for $X \in \mathbb{R}^T$, $\Pi_F \left( X - \overline{X} \right) = \left( X - \overline{X} \right)$ and that for all $k \geq K$, $W^k$ and $\Pi_{\mathbf{1}}$ commute so that:

$$\forall \tilde{k} \in \mathcal{N}, \forall k \geq K, \forall X \in \mathbb{R}^T :$$
$$W_\iota^{k+\tilde{k}-1} \ldots W_\iota^{k+1} W_\iota^k \left( X - \overline{X} \right)$$
$$= W_\iota^{k+\tilde{k}-1} \ldots W_\iota^{k+1} W_\iota^k \Pi_F \left( X - \overline{X} \right)$$
$$= \left( W_\iota^{k+\tilde{k}-1} \left( \Pi_F + \Pi_{\mathbf{1}} \right) \right) \ldots \left( W_\iota^{k+1} \left( \Pi_F + \Pi_{\mathbf{1}} \right) \right) \left( W_\iota^k \Pi_F \right) \left( X - \overline{X} \right)$$
$$= \left( W_\iota^{k+\tilde{k}-1} \Pi_F \right) \ldots \left( W_\iota^k \Pi_F \right) \left( X - \overline{X} \right).$$
$$\text{[using } \Pi_{\mathbf{1}} \Pi_F = 0 \text{]} \tag{3}$$

Therefore, an equivalent property is to show that:

$$\forall \rho \in (0,1), \exists \tilde{k} \in \mathcal{N}, \forall k \geq K, \forall X \in \mathbb{R}^T$$

$$\mathbb{E}\left[ \left\| \left( W_\iota^{k+\tilde{k}-1} \Pi_F \right) \ldots \left( W_\iota^k \Pi_F \right) \left( X - \overline{X} \right) \right\|^2 \right] \leq (1-\rho)^2 \left\| X - \overline{X} \right\|^2$$

By abuse of notation, let us omit the subscript $\iota$ index in the proof since the reasoning is parameter-independent. Additionally, we note that $W^{(k:k+\tilde{k}-1)}$ is the product of $\tilde{k}$ i.i.d random stochastic matrices that are asymmetric and not doubly-stochastic; this makes the analysis to be more involved. Let $W$ be an i.i.d copy of them.

The sketch of the proof is the following: first, we look at the expected value and the variance of the matrix $W$. Then we will use concentration inequalities to draw a result on the spectral norm of the product $W^{(k:k+\tilde{k}-1)}$.

As $\mathbb{E}[W]$ is a stochastic matrix, we start by computing the expected values of the random elements. We remark that $(\alpha^{j,k-k_{ji},i})$ and $(R^{i,k})_{i \in [n], k \geq K}$, for all $i, j \in [n], j \neq i, k \geq K$, are identically distributed as well. Moreover, we observe that $R^{i,k} \sim \text{Bin}(n-1, \frac{J}{n-1})$. Therefore,

$$\mathbb{E}\left[ \frac{1}{1+R^{i,k}} \right] = \sum_{k=0}^{n-1} \frac{1}{k+1} \binom{n-1}{k} \left( \frac{J}{n-1} \right)^k \left( 1 - \frac{J}{n-1} \right)^{n-1-k}$$

$$= \frac{n-1}{Jn} \sum_{k=0}^{n-1} \binom{n}{k+1} \left( \frac{J}{n-1} \right)^{k+1} \left( 1 - \frac{J}{n-1} \right)^{n-1-k}$$

$$= \frac{n-1}{Jn} \left( 1 - \left( 1 - \frac{J}{n-1} \right)^n \right) = \alpha_{(1)}.$$

Then using $R^{i,k} = \mathbb{E}\left[R^{i,k}|R^{i,k}\right] = \sum\limits_{1 \le j' \le n, j' \ne i} \mathbb{E}\left[\mathbf{1}(A^{j',k-k_{j'i},i})|R^{i,k}\right] =$

$(n-1)\mathbb{E}\left[\mathbf{1}(A^{j,k-k_{ji},i}|R^{i,k}\right]$, we deduce that $\mathbb{E}\left[\alpha^{j,k-k_{ji},i}\right]$

$$= \mathbb{E}\left[\frac{1}{1+R^{i,k}}\mathbb{E}\left[\mathbf{1}(A^{j,k-k_{ji},i})|R^{i,k}\right]\right]$$

$$= \mathbb{E}\left[\frac{1}{1+R^{i,k}}\frac{R^{i,k}}{n-1}\right] = \frac{1}{n-1}\left(1 - \mathbb{E}\left[\frac{1}{1+R^{i,k}}\right]\right)$$

$$= \frac{1}{n-1}\left(1 - \frac{n-1}{Jn}\left(1 - \left(1 - \frac{J}{n-1}\right)^n\right)\right) = \frac{1-\alpha_{(1)}}{n-1} = \alpha$$

Now looking at $\mathbb{E}[W]\Pi_F$, since $\Pi_F = \left(\delta_{i,j}\delta_{k_i,k_j} - \frac{1}{T}\right)_{i,k_i,j,k_j}$, we can expand the elements of the product as:

$\forall i,j \in [n], \forall 1 \le k_j \le K_j$:

$$(\mathbb{E}[W]\Pi_F)_{(i,k_i),(j,k_j)} = \begin{cases} \sum_{l,k_l} \delta_{i,l}\delta_{k_i-1,k_l}\alpha_{(1)}\left(\delta_{l,j}\delta_{k_l,k_j} - \frac{1}{T}\right) \\ = \alpha_{(1)}\left(\delta_{i,j}\delta_{k_i-1,k_j} - \frac{1}{T}\right) \text{ for } 2 \le k_i \le K_i \\ \sum_{l,k_l} \delta_{k_l,k_{l_i}}\alpha\left(\delta_{l,j}\delta_{k_l,k_j} - \frac{1}{T}\right) \\ = \alpha\left(\delta_{k_{ji},k_j} - \frac{n}{T}\right) \text{ for } k_i = 1 \end{cases}$$

We can now compute the *Frobenius norm* of this matrix:

$$\sum_{i,k_i,j,k_j} (\mathbb{E}[W]\Pi_F)^2_{(i,k_i),(j,k_j)}$$

$$= \sum_{i,j}\sum_{k_j} \alpha^2\left(\delta_{k_{ji},k_j} - \frac{n}{T}\right)^2 + \sum_{i,k_i\ge 2}\sum_{j,k_j} \alpha^2_{(1)}\left(\delta_{i,j}\delta_{k_i-1,k_j} - \frac{1}{T}\right)^2$$

$$= \sum_{i,j}\alpha^2\left(\left(1 - \frac{n}{T}\right)^2 + (K_j-1)\frac{n^2}{T^2}\right) + \sum_{i,k_i\ge 2}\alpha^2_{(1)}\left(\left(1 - \frac{1}{T}\right)^2 + \frac{T-1}{T^2}\right)$$

$$= (\alpha n)^2\left(\frac{T-n}{T}\right) + \alpha^2_{(1)}\left(T - n - 2 + 2\frac{n+1}{T} - \frac{1+n}{T^2}\right)$$

$$\le (T-n)\left(\frac{(\alpha n)^2}{T} + \alpha^2_{(1)}\right) \tag{4}$$

Using Assumption 4, we have $\sum_{(i,k_i),(j,k_j)} (\mathbb{E}[W]\Pi_F)^2_{(i,k_i),(j,k_j)} < 1$, implying $\lambda_2 = \|\mathbb{E}[W]\Pi_F\| < 1$. We now compare $\|\mathbb{E}[W]\Pi_F\|^2$ and $\mathbb{E}\left[\|W\Pi_F - \mathbb{E}[W]\Pi_F\|^2\right]$. For $X \in F$, on the one hand, $\|\mathbb{E}[W]X\|^2$

$$= \sum_{i\in[n]}\left(\alpha_{(1)}X_{i,1} + \sum_{j\ne i}\alpha X_{j,k_{ji}}\right)^2 + \sum_{1\le k\le K_i-1}X^2_{i,k}$$

$$= \sum_{i\in[n]}\left(\alpha^2_{(1)} + 1\right)X^2_{i,1} + \sum_{j\ne i}\alpha^2 X^2_{j,k_{ji}}$$

$$+ 2\sum_{j\ne j'\ne i}\alpha^2 X_{j,k_{ji}}X_{j',k_{j'i}} + 2\sum_{j\ne i}\alpha_{(1)}\alpha X_{i,1}X_{j,k_{ji}} + \sum_{2\le k\le K_i-1}X^2_{i,k}.$$

On the other hand, $\mathbb{E}\left[\|(W - \mathbb{E}[W])X\|^2\right]$

$$= \mathbb{E}\left[\sum_{i\in[n]}\left(\left(W_{(i,1),(i,1)} - \alpha_{(1)}\right)X_{i,1} + \sum_{j\ne i}\left(W_{(i,1),(j,k_{ji})} - \alpha\right)X_{j,k_j}\right)^2\right]$$

$$= \sum_{i\in[n]}\mathbb{E}\left[\left(W_{(i,1),(i,1)} - \alpha_{(1)}\right)^2\right]X^2_{i,1} + \sum_{j\ne i}\mathbb{E}\left[\left(W_{(i,1),(j,k_{ji})} - \alpha\right)^2\right]X^2_{j,k_{ji}}$$

$$+ 2\sum_{j\ne j'\ne i}\mathbb{E}\left[\left(W_{(i,1),(j,k_{ji})} - \alpha\right)\left(W_{(i,1),(j',k_{j'i})} - \alpha\right)\right]X_{j,k_{ji}}X_{j',k_{j'i}}$$

$$+ 2\sum_{j\ne i}\mathbb{E}\left[\left(W_{(i,1),(i,1)} - \alpha_{(1)}\right)\left(W_{(i,1),(j,k_{ji})} - \alpha\right)\right]X_{i,1}X_{j,k_{ji}}. \tag{5}$$

Using linearity of the expectation, we individually bound all the terms that appear in Eq. (5):

$$\mathbb{E}\left[\left(W_{(i,1),(i,1)} - \alpha_{(1)}\right)^2\right] = \mathbb{E}\left[W^2_{(i,1),(i,1)}\right] \le 1 \tag{6}$$

$$\mathbb{E}\left[\left(W_{(i,1),(j,k_{ji})} - \alpha\right)^2\right] = \mathbb{E}\left[\frac{R^i}{(1+R^i)^2}\frac{1}{n-1}\right] - \alpha^2$$

$$\le \mathbb{E}\left[\frac{1}{1+R^i}\frac{1}{n-1}\right] - \alpha^2 \le \alpha(1-\alpha) \le \frac{1-\alpha}{\alpha}\alpha^2 \tag{7}$$

$$\mathbb{E}\left[\left(W_{(i,1),(j,k_{ji})} - \alpha\right)\left(W_{(i,1),(j',k_{j'i})} - \alpha\right)\right] \text{ with } j \ne j' \ne i$$

$$= \mathbb{E}\left[W_{(i,1),(j,k_{ji})}W_{(i,1),(j',k_{j'i})}\right] - \alpha^2 \tag{8}$$

Using the same line of reasoning, $\left(R^{i,k}\right)^2 = \mathbb{E}\left[\left(R^{i,k}\right)^2|R^{i,k}\right]$

$$= \sum_{j\ne j'\ne i}\mathbb{E}\left[\mathbf{1}(A^{j,k-k_{ji},i})\mathbf{1}(A^{j',k-k_{j'i},i})|R^{i,k}\right]$$

$$+ \sum_{j\ne i}\mathbb{E}\left[\mathbf{1}(A^{j,k-k_{ji},i})^2|R^{i,k}\right]$$

$$= (n-1)(n-2)\mathbb{E}\left[\mathbf{1}(A^{j,k-k_{ji},i}\mathbf{1}(A^{j',k-k_{j'i},i})|R^{i,k}\right]$$

$$+ n\mathbb{E}\left[\mathbf{1}(A^{j,k-k_{ji},i})|R^{i,k}\right]$$

Thus, we get $\mathbb{E}\left[\left(W_{(i,1),(j,k_{ji})} - \alpha\right)\left(W_{(i,1),(j',k_{j'i})} - \alpha\right)\right]$

$$= \mathbb{E}\left[\frac{1}{(n-1)(n-2)}\frac{(R^i)^2 - R^i}{(1+R^i)^2}\right] - \alpha^2$$

$$\le \frac{\alpha}{n-2} - \alpha^2 \le \alpha^2. \tag{9}$$

As $\alpha$ is an increasing function of $J$, maximum value for the RHS in Eq. (9) is $1/n$ for $J = n-1$. Finally, we have:

$$\mathbb{E}\left[\left(W_{(i,1),(i,1)} - \alpha_{(1)}\right)\left(W_{(i,1),(j,k_{ji})} - \alpha\right)\right]$$

$$= \mathbb{E}\left[\frac{R^i}{(1+R^i)^2(n-1)}\right] - \alpha\alpha_{(1)}$$

$$\le \frac{\alpha_{(1)}(1-\alpha_{(1)})}{n-1} \le \frac{1-\alpha_{(1)}}{\alpha_{(1)}}\alpha\alpha_{(1)} \tag{10}$$

Combining Eq. (6) to (10), we obtain the overall upper bound as:

$$\mathbb{E}\left[\|(W - \mathbb{E}[W])X\|^2\right] \le \max\left(\frac{1-\alpha_{(1)}}{\alpha_{(1)}}, \frac{1-\alpha}{\alpha}\right)\|\mathbb{E}[W]X\|^2$$

Moreover, using the fact that $\alpha \le \alpha_{(1)}$ and that $x \mapsto \frac{1-x}{x}$ is decreasing, we have $\mathbb{E}\left[\|W\Pi_F - \mathbb{E}[W\Pi_F]\|^2\right] \le \frac{1-\alpha}{\alpha}\|\mathbb{E}[W\Pi_F]\|^2$. Hence, applying Corr. 5.4. from [25], we get that for all $\tilde{k} \in \mathcal{N}, k \ge K$:

$$\mathbb{E}\left[\left\|\left(W_t^{k+\tilde{k}-1}\Pi_F\right)\dots\left(W_t^k\Pi_F\right)\right\|\right]$$

$$\le \exp\left\{\sqrt{\tilde{k}}\sqrt{2\log(T)\frac{1-\alpha}{\alpha}} + \tilde{k}\log(\lambda_2)\right\}.$$

Let $0 < \rho < 1$. Then for

$$k_\rho \geq \left( \frac{\sqrt{2\log(T)\frac{1-\alpha}{\alpha}} + \sqrt{2\log(T)\frac{1-\alpha}{\alpha} + 8\log(\lambda_2)\log(1-\rho)}}{2|\log(\lambda_2)|} \right)^2,$$

for every $k \geq K$ and $X \in \mathbb{R}^T$, we finally have:

$$\mathbb{E}\left[ \left\| \left( W_t^{k+\tilde{k}-1}\Pi_F \right) \dots \left( W_t^k \Pi_F \right) \left( X - \overline{X} \right) \right\|^2 \right]$$

$$\leq \mathbb{E}\left[ \left\| \left( W_t^{k+\tilde{k}-1}\Pi_F \right) \dots \left( W_t^k \Pi_F \right) \right\|^2 \left\| X - \overline{X} \right\|^2 \right]$$

$$\leq (1-\rho)^2 \left\| X - \overline{X} \right\|^2, \text{ concluding the proof of Lem. 2.}$$

Now we proceed to prove the theoretical convergence guarantees of DivShare.

**Proof Theorem 1.** Using Lem. 2 and under Assumptions 1 to 4, we apply Theorem 4.2 from [17] and have:

$$\mathbb{E}\left[ \frac{1}{\tilde{k}} \sum_{k<\tilde{k}} \left\| \nabla F\left( \overline{X^k} \right) \right\|^2 \right]$$

$$= O\left( \frac{L\Delta \left( \frac{1}{\sqrt{n}} + \frac{ek_\rho}{(e-1)\rho} \right)}{\tilde{k}n} + \left( \frac{L\Delta \left( \sigma^2 + \zeta^2 \right)}{\tilde{k}} \right)^{\frac{1}{2}} \right.$$

$$\left. + \left( \frac{nL\Delta \sqrt{\sigma^2 \frac{ek_\rho}{(e-1)\rho} + \zeta^2 \left( \frac{ek_\rho}{(e-1)\rho} \right)^2}}{\tilde{k}} \right)^{\frac{2}{3}} \right).$$

In order to get the best bound, we reduce to the following optimization problem:

$$\min_{0<\rho<1} \frac{ek_\rho}{(e-1)\rho}$$

$$= \frac{e}{e-1} \frac{1}{\rho} \left( \frac{\sqrt{2\log(T)\frac{1-\alpha}{\alpha}} + \sqrt{2\log(T)\frac{1-\alpha}{\alpha} + 8\log(\lambda_2)\log(1-\rho)}}{2|\log(\lambda_2)|} \right)^2$$

We have:

$$\phi(\rho)$$

$$= \frac{e}{e-1} \frac{1}{\rho} \left( \frac{\sqrt{2\log(T)\frac{1-\alpha}{\alpha}} + \sqrt{2\log(T)\frac{1-\alpha}{\alpha} + 8\log(\lambda_2)\log(1-\rho)}}{2|\log(\lambda_2)|} \right)^2$$

$$\leq \frac{e}{2(e-1)|\log(\lambda_2)|^2} \frac{1}{\rho} \left( 2\log(T)\frac{1-\alpha}{\alpha} + 2\log(T)\frac{1-\alpha}{\alpha} \right.$$

$$\left. + 8\log(\lambda_2)\log(1-\rho) \right)^2$$

$$\leq \frac{2e\log(T)(1-\alpha)}{(e-1)\alpha|\log(\lambda_2)|^2} \left( \frac{1}{\rho} + \frac{2\alpha\log(\lambda_2)}{\log(T)(1-\alpha)} \frac{\log(1-\rho)}{\rho} \right)$$

The function $\rho \mapsto \frac{1}{\rho} - a\frac{\log(1-\rho)}{\rho}$ with $a > 0$ has a minimum on $\rho$ such as $\frac{\rho}{1-\rho} + \log(1-\rho) = \frac{1}{a}$.

Choosing to evaluate in $\rho = \frac{1}{1+a}$ with $a = \frac{2\alpha|\log(\lambda_2)|}{\log(T)(1-\alpha)}$ and using

$\log\left(1 + \frac{1}{a}\right) \leq \frac{1}{a}$:

$$\min_{0<\rho<1} \phi(\rho) \leq \frac{4e}{(e-1)|\log(\lambda_2)|a}(1 + 2a) \leq \frac{4e}{(e-1)|\log(\lambda_2)|}\left(\frac{1}{a} + 2\right)$$

$$\leq \frac{8e}{(e-1)} \frac{2\alpha|\log(\lambda_2)| + (1-\alpha)\log(T)}{2\alpha|\log(\lambda_2)|^2}$$

$$\leq \frac{8e}{(e-1)} \frac{\alpha|\log(\lambda_2)| + (1-\alpha)\log(T)}{\alpha|\log(\lambda_2)|^2}$$

which concludes the proof. $\square$

## G  Discussion on Assumption 4

Assumption 4 establishes a link between the effects of straggling and the communication rate in the network. In this section, we analyze the asymptotic properties of this assumption to confirm that DivShare under Assumption 4 is adaptable to a wide range of real-world scenarios. Assumption 4 can be rewritten as:

$$T \leq \hat{T} \text{ where}$$

$$\hat{T} = \frac{1}{2\alpha_{(1)}^2}\left( n\alpha_{(1)}^2 + 1 - (n\alpha)^2 + \sqrt{\left(n\alpha_{(1)}^2 + 1 - (n\alpha)^2\right)^2 + 4\alpha^2\alpha_{(1)}^2 n^3} \right).$$

**Full communication.** For $J = n - 1$, we get:

$$\hat{T} - n = n^{\frac{3}{2}}\sqrt{1 + \frac{1}{4n}} - \frac{n}{2}$$

Thus the maximum average straggling per node goes to infinity as the system scales as:

$$\frac{\hat{T} - n}{n} = \sqrt{n} - \frac{1}{2} + \frac{1}{2\sqrt{n}} + O\left(\frac{1}{n\sqrt{n}}\right)$$

**Partial communication.** For $J = \log(n)$, a parameter chosen in other works on random topology [8, 12] and in our experimental setup, we get: $\hat{T} - n \sim \log(n)^2$. In other words, depending on the communication rate chosen, if $T - n = o\left(\hat{T} - n\right)$, the coefficients of the numerators of the second and third terms in Th. 1 go to 0 as the system scales, enabling speed-ups and better convergence.

