# OpenReview forum: "Boosting Asynchronous Decentralized Learning with Model Fragmentation"
_ACM.org/TheWebConf/2025/Conference — WWW 2025 Oral_

### Official Review · Reviewer_4Mru · 2024-11-30

**Novelty:** 5
**Technical Quality:** 4

**Review:**

**Summary**

The manuscript proposes a novel approach to improving asynchronous decentralized learning by employing model fragmentation. In decentralized learning systems, models are typically trained across multiple nodes, often facing challenges like communication overhead and slow convergence due to heterogeneous computing and communication resources. The authors introduce model fragmentation as a strategy to address these issues. (1) Specifically, they suggest dividing the global model into smaller, independent fragments that can be updated asynchronously by different nodes, thus reducing the need for frequent global synchronizations. (2) The paper also emphasizes how fragmentation can help balance computational loads and mitigate communication bottlenecks in heterogeneous data distributions. (3) Experimental results suggest that this fragmentation method can lead to faster convergence times and improved performance for decentralized systems, especially in scenarios with large straggling factors.

---
**Strengths:**

**[S1]** The proposed algorithm uses the ideas from asynchronous decentralized learning, which is well applied in Web systems.

**[S2]** In the theory part, it tries to analyze the convergence guarantees of DivShare from the perspective of the global rounds.

**[S3]** The sensitive results in Section 5.3 show good performance.

**[S4]** The problem is clearly and succinctly articulated.

---
**Weaknesses:**

**[W1]** **Over-simplification of the model fragmentation strategy.** While the paper aims to address the issue of stragglers in decentralized learning scenarios by partitioning models into different parameter subsets for transmission, it fails to specify how the model should be fragmented, which model is being fragmented, or the exact process for partitioning the model into $\left \lceil \frac{1}{\Omega }  \right \rceil $ pieces. This lack of detail hinders a clear understanding of the method.

**[W2]** **Lack of detailed analysis of communication overhead.** While the manuscript emphasizes the reduction of communication overhead, there is limited discussion of how the fragmentation strategy impacts communication. The effects of latency and network variability on asynchronous updates are not sufficiently examined.

**[W3]** **Limited application in heterogeneous environments.** The proposed method may struggle to scale effectively in highly heterogeneous environments, where nodes have varying computational power and communication capabilities. The paper does not address how the fragmentation method adapts to these challenges.

**[W4]** **Lack of proof in convergence guarantees** While the authors suggest that fragmentation can speed up convergence, there is insufficient mathematical justification or theoretical analysis to support these claims. The paper would benefit from a more rigorous investigation of convergence rates under fragmentation.

**Questions:**

---
I would be willing to revise my scores if the following main questions are addressed.

1. **(Main)** The paper does not specify the model backbone used in the experiments, such as whether VGG, ResNet, or LeNet was employed. Furthermore, the method of model fragmentation remains unclear. It would be beneficial to include details on which layers or components of the model are split and provide the rationale for selecting those specific parts for partitioning.

2. **(Main)** In decentralized learning, it is essential to account for the heterogeneity among nodes and personalize the communication rounds, factors that appear to be overlooked in the current paper (random communication in *line 428*). Incorporating the Novel Client-Communication Weight Selection approach in SWIFT [1] could help personalize parameter transmission between nodes.

3. **(Main)** The paper reports a test accuracy of only 70% on CIFAR-10 (*Figure 5,6*). The authors should provide an explanation for this relatively low performance.

4. **(Main)** Could the authors provide proof of the convergence rate, similar to the approach used in [1]?

5. **(Main)** Further demonstrating the advantages of this method over the SWIFT approach [1]—either through theoretical proof or additional experimental analysis (like *Figure 5*)—would significantly strengthen the paper's contributions.

6. In all experiments, the performance of the SWIFT and AD-PSGD methods was notably similar. The authors may have overlooked key features of the SWIFT method [1], such as Wait-Free Transmission and the use of the Communication Weight Matrix to manage both regular and straggler nodes. A thorough comparison of convergence rates and communication efficiency between the proposed method and SWIFT is essential to fully evaluate the advantages and limitations of the proposed approach.

[1] Bornstein M, Rabbani T, Wang E Z, et al. SWIFT: Rapid Decentralized Federated Learning via Wait-Free Model Communication[C]//The Eleventh International Conference on Learning Representations.

**Reviewer Confidence:**

3: The reviewer is confident but not certain that the evaluation is correct

**Scope:**

4: The work is relevant to the Web and to the track, and is of broad interest to the community

---

### Official Review · Reviewer_e3GD · 2024-12-01

**Novelty:** 4
**Technical Quality:** 6

**Review:**

This paper introduces **DivShare**, an algorithm designed to enhance asynchronous decentralized learning by mitigating the impact of communication stragglers. The proposed method utilizes **model fragmentation**, a strategy where nodes fragment their models into smaller subsets and share them independently with random sets of nodes. This approach improves bandwidth utilization and enables even slower nodes to contribute partially, accelerating model convergence and improving overall performance.

Pros:
1. Comprehensive convergence proofs regarding the algorithm.
2. Experiments with CIFAR-10 and MovieLens datasets and real-world network simulations show its utility.
3. The algorithm scales well with increasing heterogeneity and straggler severity.

Cons:
1. The assumption that nodes have similar computation capabilities limits the algorithm's generality.
2. Increased message frequency, though justified, could present scalability issues in extreme network scenarios.
3. The datasets used in this paper, CIFAR-10 and MovieLens, are kind of out of date.

**Questions:**

1. Could adaptive fragmentation strategies (e.g., dynamic fragment sizing based on network conditions) further improve performance?
2. Are there plans to validate DivShare in larger-scale deployments or with additional datasets beyond CIFAR-10 and MovieLens?
3. How does DivShare handle scenarios where straggler nodes frequently join and leave the network?

**Reviewer Confidence:**

4: The reviewer is certain that the evaluation is correct and very familiar with the relevant literature

**Scope:**

4: The work is relevant to the Web and to the track, and is of broad interest to the community

---

### Official Review · Reviewer_d9KT · 2024-12-02

**Novelty:** 5
**Technical Quality:** 5

**Review:**

The introduction of the DivShare algorithm and the concept of model fragmentation is a novel contribution to the field. The paper effectively distinguishes itself from existing methods by proposing a randomized communication strategy. The methodology is clearly articulated, and the experiments are well-designed, providing a solid basis for the claims made. The writing is generally clear, with appropriate use of technical language. However, some sections could benefit from additional explanations or examples to enhance understanding, particularly for readers less familiar with decentralized learning concepts.

Pros:

- Innovative approach to model fragmentation and communication.

- Comprehensive experimental evaluation across multiple datasets.

- Formal proof of convergence adds rigor to the claims.

Cons:

- Some technical details may be challenging for a broader audience.

- The impact of network characteristics on performance could be explored in more depth.

**Questions:**

- How does DivShare compare to other state-of-the-art methods in terms of scalability and adaptability to different network conditions, particularly in highly heterogeneous environments?
- Could you provide more insights into the computational/communication overhead introduced by the model fragmentation process?
- Have you considered the implications of data privacy and security in the context of model fragmentation and sharing?

**Reviewer Confidence:**

2: The reviewer is willing to defend the evaluation, but it is likely that the reviewer did not understand parts of the paper

**Scope:**

2: The connection to the Web is incidental, e.g., use of Web data or API

---

### Official Review · Reviewer_RJ7W · 2024-12-02

**Novelty:** 5
**Technical Quality:** 7

**Review:**

- Introduction: Decentralized learning (DL) is a collaborative learning framework that allows nodes on the web" -> why only on the web? DL can we applied also in IoT.

- What is an example case of DL on the web and what is a representative percentage of  communication stranglers? In Figure 1 half of the nodes have 5x slower network speeds than others. This number though sounds a bit arbitrary

- Authors provide both a formal proof of convergence and an evaluation of their approach based on both simulations and real world scenarios.

- The assumption of static participation is quite restricting (and possibly impractical?) considering the flexible nature of DL where usually there is churn of users being unexpectedly online or offline across different rounds.

- Is there a maximum threshold with regards to communication delays? Can a node delay for hours? what if the node is network partitioned and manage to reconnect after a day? How Divshare can filter out their outdated model?

- Line 230: The authors acknowledge the challenges of asynchronous DL algorithms yet they do not discuss anywhere in the paper how these challenges could be mitigated since in Divshare faster devices can clearly introduce bias.

- Line 272: "For clarity and presentation, we assume the nodes have comparable computation infrastructure that allows them to compute at the same speed". What if the nodes are nearby (no network latency) but there is delay due to their heterogeneity of computation power (some devices are less powerful than others)? Would Divshare still work?

- I like the fact that authors compare their approach with state-of-the-art asynchronous algorithms

- Related work is way too short considering the abundance of existing works in the area of DL and peer-to-peer learning the past few years.

**Questions:**

- How DivShare can mitigate the bias of the faster nodes?

- Is there a maximum latency a node can suffer from in order to be able to participate in the DL without polluting with their outdated model? Is there a way for the consortium of nodes to enforce some policy for very slow nodes?

**Reviewer Confidence:**

2: The reviewer is willing to defend the evaluation, but it is likely that the reviewer did not understand parts of the paper

**Scope:**

3: The work is somewhat relevant to the Web and to the track, and is of narrow interest to a sub-community